# Responsiveness and interpretability of the pain subscale of the Knee and Hip Osteoarthritis Outcome Scale (KOOS and HOOS) in osteoarthritis patients according to COSMIN guidelines

**Wietske Rienstra**[1,2]*, **Martin Stevens**[2], **Tim Blikman**[1,2], **Sjoerd K. Bulstra**[2], **Inge van den Akker-Scheek**[2]

1 Department of Rehabilitation, University Medical Center Groningen, University of Groningen, Groningen, The Netherlands, 2 Department of Orthopaedics, University Medical Center Groningen, University of Groningen, Groningen, The Netherlands

* w.rienstra@revalidatie-friesland.nl

**Data Availability Statement:** All relevant data are within the manuscript and its Supporting

## Abstract

### Background

The pain subscales of the Knee and Hip Osteoarthritis Outcome Scores (KOOS and HOOS) are among the most frequently applied, patient reported outcomes to assess pain in osteoarthritis patients and evaluation of the results after Total Knee Arthroplasty (TKA) and Total Hip Arthroplasty (THA). For the evaluation of change over time it is essential to know the responsiveness and interpretability of these measurement instruments. Aim of this study is to investigate responsiveness and interpretability of the KOOS and HOOS pain sub-scales in patients with knee or hip OA and patients after TKA and THA as recommended by COSMIN guidelines. COSMIN stands for COnsensus based Standards for the selection of health Measurement Instruments. COSMIN recommends methods for assessing respon-siveness similar to those assessing validity, using extensive hypothesis testing to assess criterion validity and construct validity of the change score.

### Design

This clinimetric study was conducted using data obtained from the Duloxetine in OsteoAr-thritis (DOA) trial. Primary knee or hip osteoarthritis patients were included. During the study, half of the participants received pre-operative targeted treatment with duloxetine, and all participants received TKA or THA. Patients filled out a set of patient-reported outcomes at several time points.

### Methods

Using the criterion validity approach the change scores of the KOOS and HOOS pain sub-scales directly after duloxetine treatment but before TKA and THA were correlated to the

information file. The underlying dataset on which all results were founded is included within the Supporting information file. This file is mentioned in the manuscript at the start of the Results section (line 299).

**Funding:** This work was supported by the Dutch Arthritis Foundation, https://reumanederland.nl; (grant number BP 12-3-401) The funders had no role in study design, data collection and analysis, decision to publish, or preparation of the manuscript.

**Competing interests:** The authors have declared that no competing interests exist.

Patient Global Improvement anchor-question (PGI-I). Receiver Operating Characteristic curves (ROC curves) were obtained. Using the construct validity approach, hypothesis testing was conducted investigating the correlation between change scores in the KOOS and HOOS pain subscale with change scores in other questionnaires six months after TKA and THA. For interpretability, an anchor-based approach was used to consider the Minimally Important Change (MIC) of the KOOS and HOOS pain subscale. We compared the outcomes after duloxetine treatment and six months after TKA and THA in order to investigate any response shift.

## Results

Ninety-three participants (53 knee patients and 41 hip patients) were included. Mean change was 4.3 and 4.6 points after conservative treatment for knee and hip OA patients respectively and 31.7 and 48.8 points after TKA and THA respectively. The KOOS and HOOS pain subscales both showed a gradual increase in change scores over the different categories of improvement on the PGI-I, with an Area Under the Curve of 0.72 (95% CI 0.527–0.921) and 0.79 (95% CI 0.588–0.983) respectively. Of the predefined hypotheses, 69% were confirmed for both subscales. The MICs were between 12.2 to 37.9 for the KOOS pain subscale, and between 11.8 to 48.6 for the HOOS pain subscale, depending on whether the PGI-I was administered after conservative treatment, or six months after TKA and THA.

## Conclusions

This study endorses the responsiveness of the KOOS and HOOS pain subscales in patients with knee or hip OA and patients after TKA and THA based on construct and criterion validity approaches. The KOOS pain subscale might be able to detect the MIC at an individual level after arthroplasty, but both the KOOS and HOOS pain subscales were not able to do so after conservative treatment. This study is the first to report a considerable response shift in MIC of the KOOS and HOOS pain subscales. This should be taken into consideration when evaluating MIC of the KOOS and HOOS pain subscale after conservative versus operative treatment. Future research should present more reference data regarding MIC scores after different treatments.

## Introduction

Osteoarthritis (OA), one of the leading causes of disability worldwide, constitutes a heterogeneous syndrome that is evaluated predominantly through clinical symptoms. Pain, its main symptom, is assessed mostly using patient reported outcomes [1–5]. Longitudinal studies form an important part of clinical research regarding pain in OA, for example to assess disease progression over time, and to assess effects of different treatment options. For the evaluation of change over time it is essential to know the responsiveness and interpretability of the measurement instruments used. COSMIN (COnsensus based Standards for the selection of health Measurement INstruments) defines responsiveness as 'the ability of an instrument to detect change over time in the construct to be measured', and interpretability as 'the degree to which one can assign qualitative meaning (in other words: clinical or commonly understood connotations) to an instrument's quantitative change scores [6, 7]. Interpretability is not so much a

measurement property of an instrument, but it is paramount for the application of the instrument in clinical research.

The pain subscales of the Knee and Hip Osteoarthritis Outcome Scores (KOOS and HOOS) are among the most frequently applied, patient reported outcomes to assess pain in OA patients and after Total Knee Arthroplasty (TKA) and Total Hip Arthroplasty (THA) [2–4, 8]. In a recent systematic review and meta-analysis of the measurement properties of the KOOS it turns out that evidence of responsiveness almost solely originates from studies reporting effect sizes (ES) and standardized response means (SRM) after various interventions [3]. However, according to the COSMIN criteria, ES and SRM provide very limited evidence for responsiveness. In fact, these distribution-based approaches are considered inappropriate measures of responsiveness as they express the magnitude of change (for example after an intervention) but not whether the change is valid [6]. COSMIN recommends methods for assessing responsiveness similar to those assessing validity, using extensive hypothesis testing to assess criterion validity and construct validity of the change score [6]. Only if the ES are incorporated into an approach using a priori defined hypotheses its use is found acceptable in responsiveness studies. Responsiveness analysis is a continuous process of accumulating evidence, depending on study population, context, and properties of the instruments used for comparison [6]. A recent meta-analysis shows only 2/17 studies regarding responsiveness of the KOOS used pre-defined hypothesis according to COSMIN criteria, both studies did not assess primary OA patients [3, 9, 10]. To our knowledge, only one study assessing SRM in patients after THA is available, without hypothesis testing or following other COSMIN criteria [11].

Regarding interpretability of change scores in clinical practice the Minimally Important Change (MIC) and the Smallest Detectable Change (SDC) are especially informative. In literature, reports of MIC of the KOOS and HOOS pain subscale are scarce.

Therefore, further evaluation of the responsiveness and interpretability of the KOOS and HOOS pain subscales using COSMIN recommended methods is needed [3]. Consequently, the aims of this study are to investigate responsiveness and interpretability of the KOOS and HOOS pain subscales in patients with knee or hip OA as well as patients after TKA and THA as recommended by COSMIN guidelines.

## Methods

This clinimetric study was conducted using data obtained from the Duloxetine in OsteoArthritis (DOA) study [12]. This is a prospective, randomized, clinical trial assessing the effect of pre-operative duloxetine treatment in sensitized knee and hip OA patients on chronic residual pain after TKA and THA. Participating hospitals were University Medical Center Groningen, Martini Hospital Groningen, and Medical Center Leeuwarden. The study was approved by the Medical Ethics Committee of University Medical Center Groningen (2014/ 087) and the procedures followed were in accordance with the ethical standards of the responsible committee on human experimentation and with the Helsinki Declaration of 1975, as revised in 2000. Written informed consent was obtained from all participants prior to entering the study.

Details regarding the DOA study were published earlier [12]. Patients were recruited between December 2014 and June 2018. Patients who were included all suffered from primary knee or hip OA and reported a possible or likely neuropathic pain phenotype as assessed with the modified PainDetect Questionnaire (mPDQ), indicating signs of sensitization. During the study, half of the participants received pre-operative targeted treatment with duloxetine, and all participants received TKA or THA following the local protocol. Authors (WR and TB) had access to information that could identify individual participants during or after data collection.

In the course of the study period, patients filled out a set of patient-reported outcomes at several time points [12].

## Measurement instruments

The outcome measures included the KOOS and HOOS, modified PainDetect Questionnaire (mPDQ), Visual Analoque Scales for pain (VAS), the RAND-36 Health Survey (RAND-36), Hospital Anxiety and Depression Scale (HADS), and Pain Catastrophizing Scale (PCS). At the follow-up time points patients also filled out a Patient reported Global Improvement (PGI) score.

**Knee and Hip Osteoarthritis Outcome Scores (KOOS and HOOS).** The KOOS and the HOOS are self-administered; disease-specific questionnaires designed to assess patients' opinion about their knee or hip symptoms and associated problems. Both scores consist of five subscales: Pain, other Symptoms, Activities of Daily Living (ADL), Sport and Recreational function, and hip/knee related Quality of Life (QOL). Answers are given on a 0–4 Likert scale. Missing items in the KOOS and HOOS are imputed according to the KOOS and HOOS manual. For the pain subscales a normalized 0–100 score is calculated. These 0–100 scores were transformed so that 0 represents extreme pain and 100 represents no pain. Several clinimetric properties of the KOOS and HOOS have been assessed quite extensively in previous literature, reporting good reliability and validity, also of the Dutch version of the KOOS and HOOS [3, 9, 11, 13–15].

**Dutch modified PainDETECT Questionnaire (mPDQ).** The mPDQ is a self-administered questionnaire consisting of 12 items on neuropathic pain symptoms in the left/right knee or hip during the past week. The first item concerns the presence of pain radiation using a body map. The second item concerns pain patterns, where patients have to choose between four figures representing distinctly described (and visually illustrated) pain patterns. Seven items concern pain quality on a 0–5 Likert scale, 0 representing 'never' and 5 representing 'very strongly'. These items concern burning sensation, tingling or prickling sensation, pain at light touch, sudden pain attacks, pain at cold or warm stimulus, numbness and pain at light pressure, respectively. The total score ranges from -1 to 38 points. Analogously to the original PainDETECT Questionnaire, a score of ≤12 indicates a nociceptive pain profile, a score of 13–18 a possible neuropathic pain profile, and a score ≥19 a likely neuropathic pain profile [16].

**Visual Analogue Scale pain (VAS pain).** Visual Analogue Scales (VAS) are widely used to measure pain. Patients place a marking on a 100-mm horizontal line that represents their pain. Patients were asked to note their present pain status and their mean pain status over the last week; at rest (VAS-R: defined as pain in rest while sitting, standing or lying down) and during movement (VAS-M defined as pain during regular walking).

**RAND-36 Health Survey.** The RAND-36 Health Survey (RAND-36) is a widely used self-administered, generic health status questionnaire that assesses quality of life and well-being [17]. It contains 36 questions and standardised response choices. These questions are divided into eight different subscales: physical functioning, role limitations due to physical health problems, role limitations due to emotional problems, social functioning, vitality, mental health, bodily pain, and general health perceptions. The Mental and Physical Component Summary (MCS and PCS) can be calculated combining scores from different subscales. For this study the subscales physical functioning, bodily pain, and MCS were used. All scores are converted to a 0-to-100 scale, with a higher score indicating higher levels of functioning or well-being. The Dutch version of the RAND-36 is considered a highly reliable instrument with satisfactory validity [17, 18].

**Hospital Anxiety and Depression Scale (HADS).**    The HADS is a screening questionnaire regarding anxiety and depressive symptoms. The HADS contains two 7-items subscales: anxiety and depression. Each subscale is a 4-point Likert scale, ranging from 0–3 according to the severity of experienced distress. The reliability and dimensional structure seem to be stable for the Dutch version, across medical setting and age groups [19].

**Pain Catastrophizing Scale (PCS).**    The PCS is a questionnaire developed to identify catastrophic thoughts or feelings in relation to experienced pain [20]. The questionnaire consists of 13 items that reflect on past painful experiences. The items are scored on a 0 (not at all)—4 (all the time) Likert scale. The total score ranges from 0 to 52; higher scores reflect more catastrophizing thoughts or feelings. The test-retest reliability and internal consistency of the Dutch version of the PCS are sufficient and stable [21]. Construct and concurrent validity seem sufficient [20].

**Patient Global Impression of Improvement (PGI-I).**    The PGI-I scale is a one-item questionnaire that can measure the patients perceived change in a clinical important outcome. It is a 7-point Likert scale that ranges from "very much worse" to "very much improved". It is derived from the clinical global impression scale [22]. The question that was asked patients was 'the extent to which their hip/knee complaints had changed compared to the start of the study, when they were placed on the waiting list for THA/TKA'. The PGI-I questionnaire was previously used in several other studies concerning patients with musculoskeletal pain [23–25].

## Procedure

First, the distributions of baseline and follow-up KOOS and HOOS pain subscales were explored. When assessing responsiveness and interpretability of longitudinal data, floor and ceiling effects can pose problems. If a large proportion of patients score on the highest or the lowest side of the measurement scale, it can affect the ability of detection of relevant changes (depending on the direction of interest). Floor and ceiling effects can occur if more than 15% of the patients achieve either the lowest or the highest score [6].

**Criterion validity of change scores.**    Using the criterion validity approach (comparison to a gold standard) it is important to be able to assume that a portion of the study population has changed and another portion has not changed [6]. Therefore, for this approach, we used the change scores of the KOOS and HOOS pain subscales directly after duloxetine treatment but before TKA and THA (which was nine weeks after baseline). As half of the participants had received pre-operative duloxetine treatment at this time point, and the other half had received care as usual (for details see previous studies regarding DOA study), it is likely that only a portion of the study population changed. Based on literature, duloxetine treatment seems to have a modest positive effect on OA pain, especially in patients showing signs of sensitization [26, 27]. The changes in KOOS and HOOS pain subscales were correlated to the PGI-I anchor question. The use of a Global Rating Scale, such as the PGI-I, as a 'gold standard' is considered suitable in the criterion approach of responsiveness of patient reported outcomes [6]. Keeping in mind a certain degree of measurement error in both the pain subscale of the KOOS and HOOS, and in the PGI-I, at baseline and at follow-up, we hypothesized a correlation $\geq 0.6$.

Receiver Operating Characteristic curves (ROC curves) were obtained as additional assessment of criterion validity. The PGI-I was dichotomised into a group of patients with relevant improvement (PGI-I scales 'much improved', and 'very much improved') and a group of patients with no relevant improvement (PGI-I scales 'minimally improved', 'no change', and 'minimally deteriorated'). The area under the ROC curves is considered to be a determination of the ability of the questionnaire to discriminate between patients who improved on the

PGI-I and those who did not improve on the PGI-I. An Area Under the Curve (AUC) of $\geq 0.70$ can be considered appropriate [6].

**Construct validity of change scores.** To assess responsiveness using the construct validity approach, it is necessary to assume that actual change occurs in the study population [6]. TKA and THA are among the most performed surgical procedures in orthopaedic surgery and its effectiveness is well established in patients with end-stage knee- and hip OA with considerable effect sizes [3, 28, 29]. Therefore, we used the change scores between baseline and six months after TKA and THA for the assessment of construct validity. Elaborate hypothesis testing was conducted investigating a-priori defined hypotheses regarding the correlation between change scores in the KOOS and HOOS pain subscale with change scores in other questionnaires (see Table 4). Converging and diverging validity were assessed by comparing the relationship of the change in KOOS and HOOS outcome scores with change in instruments with similar (VAS pain scales and RAND-36 pain subscales) and different constructs (mPDQ, RAND-36 Mental Component Summary, HADS, and PCS). In accordance to literature, responsiveness was considered high if >75% of the previously composed hypotheses were met, moderate if 25–50% were met and poor if <25% were met [6].

**Interpretability.** The minimal important change (MIC) is defined by COSMIN as 'the smallest change in score in the construct to be measured which patients perceive as important'. In the present study, we used an anchor-based approach to consider the MIC of the KOOS and HOOS pain subscale, based on the subcategory of patients who reported 'much improved' on the PGI-I. We compared the outcomes after conservative treatment (nine weeks after baseline), and six months after TKA and THA in order to investigate any response shift.

## Statistical analyses

Statistical analyses were performed using IBM SPSS Statistics for Windows (version 23.0, Armonk, NY: IBM Corp.). Patient characteristics were reported using descriptive statistics consisting of mean and standard deviation or median and interquartile ranges in case of non-parametric variables. Correlation was presented either using Pearson or Spearman correlation coefficients, depending on normality. According to COSMIN guidelines, criterion and construct approaches require a minimal sample size of 50 patients, but larger samples are preferred [6].

## Results

S1 Data provides the dataset underlying the results of this study.

## Demographics and descriptive statistics

One hundred and eleven patients were included in the original DOA study. Of these, the PGI-I and KOOS and HOOS pain subscales at six months after TKA and THA were available for 93 participants (53 knee patients and 41 hip patients), as a result these were included in the responsiveness analyses. Reasons for loss to follow-up were described previously [30]. Table 1. shows the baseline characteristics of the patients included in the responsiveness analyses. The patients with knee OA group consisted of more females compared to the hip OA group. Age and Body Mass Index were comparable between both groups. Most patients suffered from grade III OA on the Kellgren and Lawrence scale. Table 2. shows the mean scores and distribution at baseline as well as after conservative treatment, and six months after TKA and THA. Also, the mean change scores are presented.

**Table 1. Demographics and baseline characteristics of the study group.**

|  | Knee patients (N = 53) | Hip patients (N = 41) |
|---|---|---|
| Age (years; mean (SD)) | 64.4 (7.8) | 61.8 (9.1) |
| Female N (%) | 36 (69.2) | 23 (55.3) |
| BMI (kg/m2; mean (SD)) | 29.6 (4.9) | 27.9 (4.3) |
| Duration of pain symptoms (mos; median (IQR)) | 69.7 (24.0; 72.0) | 24.0 (12.8; 61.3) |
| ASA score (N = 52) |  |  |
| I | 14 (26.9) | 15 (36.8) |
| II | 33 (63.5) | 23 (57.9) |
| III | 5 (9.6) | 3 (5.3) |
| KL grade (N = 52) |  |  |
| II | 7 (13.5) | 13 (31.6) |
| III | 44 (84.6) | 25 (63.2) |
| IV | 1 (1.9) | 3 (5.3) |

N: number of patients; SD: Standard Deviation; BMI: Body Mass Index; IQR: Inter Quartile Range; ASA score: American Society of Anesthesiologists Physical Status Score; KL grade: Kellgren and Lawrence severity of OA scale.

## Criterion validity of change scores

After conservative treatment the KOOS and HOOS pain subscales both showed a gradual increase in change scores over the different categories of improvement on the PGI-I, as can be seen in Table 3.

Figs 1 and 2 show the ROC curves of the KOOS and HOOS pain subscales after conservative treatment. In the knee OA study group, n = 29 knee-OA patients were 'unchanged' after conservative treatment; n = 11 knee-OA patients were 'improved'. The AUC was 0.72 (95% CI 0.527–0.921). In the hip OA study group, n = 22 hip-OA patients were 'unchanged' after conservative treatment; n = 9 hip-OA patients were 'improved'. The AUC was 0.79 (95% CI 0.588–0.983).

## Construct validity of change scores

Table 4 shows a cross table of the correlations between change in KOOS and HOOS pain subscales (columns) and change scores of other outcome scores (rows) six months after TKA and THA. Our pre-defined hypotheses are presented in parentheses, 69% of the hypotheses were

**Table 2. Mean scores and change scores at baseline and at follow-up time points.**

|  | KOOS pain subscale (0–100) | HOOS pain subscale (0–100) |
|---|---|---|
| **Baseline** |  |  |
| Mean ± SD | 34.6 ± 13.4 | 34.5 ± 14.6 |
| **After treatment (9 weeks after baseline)** |  |  |
| Mean ± SD | 39.7 ± 16.3 | 38.7 ± 14.7 |
| Mean Change ± SD | 4.3 ± 10.9 | 4.6 ± 11.9 |
| **Six months after TKA and THA** |  |  |
| Mean ± SD | 67.6 ± 25.0 | 84.6 ± 19.6 |
| Mean Change ± SD | 31.7 ± 25.7 | 48.8 ± 25.9 |

KOOS: Knee Osteoarthritis Outcome Score; HOOS: Hip Osteoarthritis Outcome Score; SD: Standard Deviation

**Table 3. Change in KOOS and HOOS pain subscales after conservative treatment for the different categories of improvement on the PGI-I.**

| PGI-I | ΔKOOS pain subscale (n = 53) | | ΔHOOS pain subscale (n = 40) | |
|---|---|---|---|---|
| | N (%) | Mean ± SD | N (%) | Mean ± SD |
| Very much improved | 3 (5.7) | 18.5 ± 18.9 | 2 (5) | 20.0 ± 7.1 |
| Much improved | 8 (15.1) | 12.2 ± 11.0 | 7 (17.5) | 11.8 ± 11.5 |
| Minimally improved | 7 (13.2) | 2.4 ± 9.7 | 3 (7.5) | 6.7 ± 8.0 |
| No change | 8 (15.1) | 4.9 ± 7.1 | 9 (22.5) | 7.5 ± 10.2 |
| Minimally deteriorated | 14 (26.4) | 3.8 ± 7.4 | 10 (25) | 4.8 ± 7.6 |
| Much deteriorated | 12 (22.6) | -2.8 ± 10.7 | 8 (20) | -5.9 ± 12.2 |
| Very much deteriorated | 1 (1.9) | NA | 1 (2.5) | NA |
| $SDC_{ind} = 1.96 \cdot SEM \cdot \sqrt{2}$ | | 13.9 | | 20.0 |
| $SDC_{group}$ | | 4.9 | | 6.7 |

Δ: change; N: number; KOOS: Knee Osteoarthritis Outcome Score; HOOS: Hip Osteoarthritis Outcome Score; PGI-I: Patient Global Improvement Index; SD: Standard Deviation; SDC: Smallest Detectable Change; NA: Not Applicable.

confirmed for both KOOS and HOOS pain subscales. Confirmed hypotheses are presented in bold.

## Interpretability

Table 5 presents the change in KOOS and HOOS pain subscales six months after TKA and THA according to the different categories of PGI-I. Based on Tables 3 and 5, the MICs are

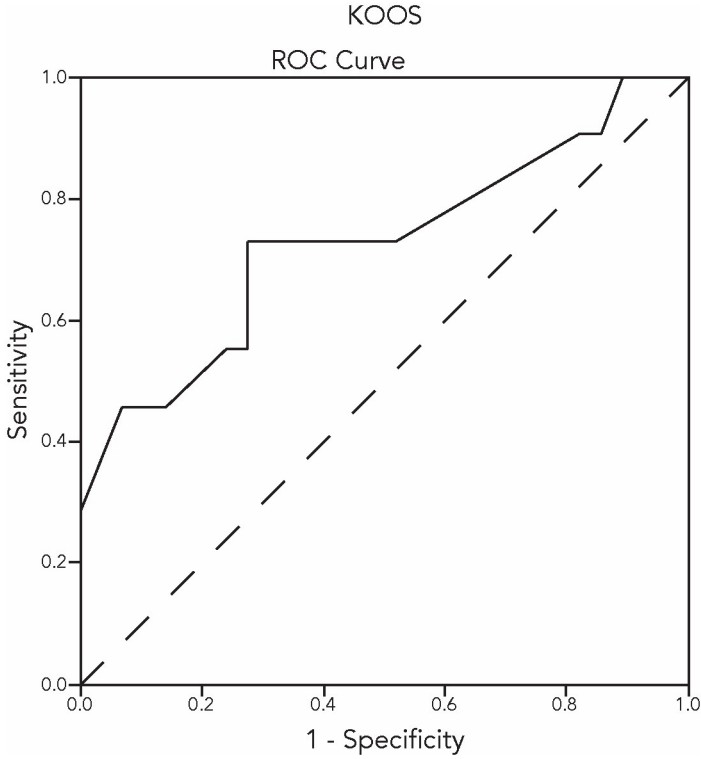

**Fig 1. The ROC curve regarding the KOOS change scores after conservative treatment.**

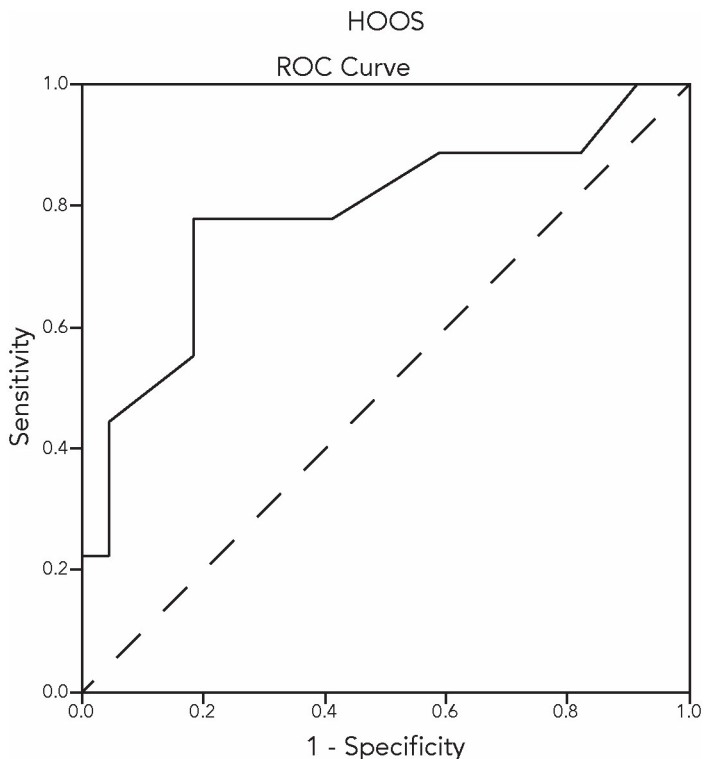

**Fig 2. The ROC curve regarding the HOOS change scores after conservative treatment.**

**Table 4. Cross table regarding correlations between change in KOOS and HOOS pain subscales and change on other outcome scores six months after TKA and THA.**

|  |  | ΔKOOS pain subscale | ΔHOOS pain subscale |
|---|---|---|---|
| Construct approach |  |  |  |
| 1 | ΔVAS rest | **0.7 (≥0.6)** | **0.6 (≥0.6)** |
| 2 | ΔVAS movement | **0.8 (≥0.6)** | **0.6 (≥0.6)** |
| 3 | ΔmPDQ | **0.7 (≥0.5)** | **0.6 (≥0.5)** |
| 4 | ΔKOOS/HOOS ADL subscale | **0.9 (≥0.7)** | **0.9 (≥0.7)** |
| 5 | ΔKOOS/HOOS QOL subscale | **0.9 (≥0.7)** | **0.7 (≥0.7)** |
| 6 | ΔKOOS/HOOS Symptoms subscale | **0.7 (≥0.7)** | **0.8 (≥0.7)** |
| 7 | ΔRand 36 physical functioning | 0.7 (0.2–0.4) | 0.6 (0.2–0.4) |
| 8 | ΔRand 36 Bodily pain | 0.7 (0.4–0.6) | **0.6 (0.4–0.6)** |
| 9 | ΔMental scale Rand-36 | **0.3 (≤0.3)** | **-0.1 (≤0.3)** |
| 10 | ΔPCS | 0.6 (0.2–0.4) | 0.6 (0.2–0.4) |
| 11 | ΔHADS depression subscore | **0.2 (0.2–0.4)** | **0.4 (0.2–0.4)** |
| Criterion approach |  |  |  |
| 11 | Correlation with PGI-I | 0.7 (≥0.6) | 0.5 (≥0.6) |
| 12 | AUC of ROC curves | 0.89 (≥0.7) | 0.96 (≥0.7) |
| Hypotheses met |  | 9/13 (69%) | 9/13 (69%) |

Δ: change; KOOS: Knee Osteoarthritis Outcome Score; HOOS: Hip Osteoarthritis Outcome Score; VAS: Visual Analoque Scale; mPDQ: modified PainDETECT Questionnaire; PCS: Pain Catastrophizing Scale; HADS: Hospital Anxiety and Depression Scale; PGI-I: Patient Global Improvement Index; AUC: Area Under the Curve; ROC: Receiver Operating Characteristic; ADL: Activities in Daily Life; QOL: Quality Of Life.

**Table 5. Change in KOOS and HOOS pain subscales six months after TKA and THA for the different categories of improvement on the PGI-I.**

| PGI-I | ΔKOOS pain subscale (n = 52) | | ΔHOOS pain subscale (n = 41) | |
|---|---|---|---|---|
| | N (%) | Mean ± SD | N (%) | Mean ± SD |
| Very much improved | 13 (25.0) | 56.0 ± 14.0 | 23 (56.1) | 57.2 ± 20.3 |
| Much improved | 20 (38.5) | 37.9 ± 20.2 | 15 (36.6) | 48.6 ± 26.4 |
| Minimally improved | 13 (25.0) | 13.0 ± 21.5 | 3 (7.3) | 0.8 ± 13.8 |
| No change | 3 (5.8) | 11.1 ± 2.8 | 0 (0) | NA |
| Minimally deteriorated | 3 (5.8) | 0.9 ± 14.3 | 0 (0) | NA |
| Much deteriorated | 0 (0) | NA | 0 (0) | NA |
| Very much deteriorated | 0 (0) | NA | 0 (0) | NA |
| $SDC_{ind} = 1.96 \cdot SEM \cdot \sqrt{2}$ | | 5.5 | | NA |
| $SDC_{group}$ | | 3.2 | | NA |

Δ: change; N: number; KOOS: Knee Osteoarthritis Outcome Score; HOOS: Hip Osteoarthritis Outcome Score; PGI-I: Patient Global Improvement Index; SD: Standard Deviation; SDC: Smallest Detectable Change; NA: Not Applicable.

between 12.2 to 37.9 for the KOOS pain subscale, and between 11.8 to 48.6 for the HOOS pain subscale, depending on whether the PGI-I was administered after conservative treatment, or six months after TKA and THA. The Smallest Detectable Change (SDC) on individual level and group level are also presented in Tables 3 and 5 in order to be able to compare MIC with the SDC.

## Discussion

Aim of this study was to investigate responsiveness and interpretability of the KOOS and HOOS pain subscale in patients with knee or hip OA and in patients after TKA and THA following COSMIN guidelines. The results of the present study endorse the responsiveness of both subscales. When comparing changes in the KOOS and HOOS pain subscales to the PGI-I, both questionnaires showed a gradual increase in change scores over the different categories of improvement. Also, the AUC of the ROC curves were > 0.7. Considering responsiveness using hypothesis testing, 69% of predefined hypotheses were met for both knee and hip OA patients. In accordance to literature, this would implicate a moderate rate of confirmed hypotheses [6]. When looking closer at the correlations found, most correlation coefficients were higher than hypothesized. Apparently we were rather conservative in our hypotheses, which was based on the COSMIN advice to consider the fact that we were dealing with the correlation between change scores in which we had to account for measurement error in baseline and follow-up measures for both instruments [6]. The mPDQ is considered to be a fairly reliable and valid self-report instrument in patients with hip and knee OA with an individual SDC of 7.3 points for hip and knee OA patients which also prompted us to be rather conservative in our hypothesis [31, 32].

Considering interpretability, there was a considerable difference between the change scores that patients report to be minimally clinically important after conservative treatment and six months after TKA and THA. This might represent a response shift between what is considered a relevant change of OA pain after conservative treatment, and after TKA and THA. It seems that a sort of 'recalibration' takes place of the internal perception of pain relieve patients experience before and after TKA and THA. The phenomenon of response shift in knee OA patients has been reported previously in literature, but not considering the size of the MIC [33, 34]. When patients were asked six months after TKA to judge how their pain and disability was

prior to surgery they tended to judge themselves worse compared to how they rated their complaints preoperatively [33]. This was especially the case for the appraisal of pain. In other words, it is known from literature that the way patients recall their previous pain experience when they were suffering from OA seems to be affected by the change of the internal standard of measurement of pain relief. The present study adds that also the size of the MIC seems to change due to this phenomenon. This is important to take into consideration when interpreting MIC scores on the KOOS and HOOS pain subscales after conservative versus operative treatment. Different MIC scores in KOOS and HOOS pain subscales should be used depending on the treatment at hand, and future research should present more reference data regarding MIC scores after different treatments. The MIC in KOOS pain subscale after conservative treatment was 12.2 points, and after TKA 37.9 points. The MIC of the HOOS pain subscale was 11.8 after conservative treatment and 48.6 after THA. When comparing our findings to literature, a previous study reported a MIC of 12 points in KOOS pain subscale after non-surgical management in 112 patients, which is in line with the present finding of 12.2 points [35]. Another study found a MIC of 16.7 after TKA rehabilitation in 148 patients using the anchor-based method, which is considerably lower than the present finding of 37.9 [36]. We could not find comparable MIC reports in literature regarding the HOOS pain subscale.

When comparing the MIC to the SDC, the KOOS pain subscale might be able to detect MIC at individual level after TKA, but not after conservative treatment. The present study reports SDCind of 13.9 after conservative treatment and 5.5 after TKA. This means the SDC is smaller than the MIC at six months after TKA, and after conservative treatment the SDC is larger than the MIC. Therefore, the KOOS pain subscale cannot detect MIC at individual level after conservative treatment, as the standard error of measurement is too large for that compared to the relatively small MIC reported in this study. In clinical research, the study groups should be sufficiently large in order to reduce the measurement error. The HOOS pain subscale does not seem able to detect MIC at an individual level after conservative treatment, as the SDC after conservative treatment is larger than the MIC (see Table 4). Based on the current data, the SDCind six months after THA could not be calculated in a reliable way due to the small number of patients that not improve after surgery, and to our knowledge there is no previous report in literature either.

Floor- and ceiling effects were not found. Baseline KOOS and HOOS pain subscale scores were at the low side of the scale and there was enough room for improvement. At six months after TKA and THA mean KOOS and HOOS pain subscales were 67.6 and 84.6 respectively, indicating that most patients were on the high end of the scale, as would be expected, however, using the >15% of participants scoring the maximum score possible as criterion (REF), there were no ceiling effects.

To our knowledge this is the first study assessing responsiveness and interpretability of the HOOS and KOOS pain subscale in patients suffering from OA using both hypothesis testing (construct validity) and anchor-based approaches (criterion validity) as recommended by COSMIN. Collins et al. report that the Pain subscale especially, and also the ADL subscale, and QOL subscale of the KOOS show the greatest relevance, room for improvement, and effect sizes in knee OA patients undergoing TKA. Collins et al. report a Smallest Detectable Change (SDC) of 26 points on individual level for the pain subscale of the KOOS in patients with OA [3]. To our knowledge, similar information is lacking in literature regarding the HOOS, increasing the relevance of the present study. However, this study has its limitations. Although the total number of patients is sufficient, the separate groups within the different categories of improvement on the PGI-I were small, limiting interpretability of some of these results. However, this is partly inevitable due to the successfulness of TKA and THA. The patient subgroups reporting 'no change' or 'minimal improvement' will probably always be relatively small.

Furthermore, we did not use an anchor question referring to the exact same construct as the pain subscale of the KOOS and HOOS. Gold standards for patient-reported questionnaires are very rare. Instead, a Global Rating Scale (GRS) assessing change perceived by the patient compared to baseline in a single question can be considered a gold standard [6]. This GRS should regard the same construct as the measurement instrument under consideration. The PGI-I asks patients to compare their present complaints to those at baseline. Considering that pain is the predominant complaint in OA patients, we think that, in the context of this study the PGI-I can be considered a good instrument for comparison.

There are some concerns when evaluating responsiveness using the construct validity approach during a clinical study in which the measurement instrument under study is also an outcome score. One must be able to assume that the study population actually changed, because the interpretation of the results can be difficult if the (lack of) measurement properties of the instrument cannot be disentangled from the (lack of) effect of the intervention. All patients in this study received a TKA and THA in the course of the study. The arthroplasty itself was not the intervention under study, and whether there would be an effect of arthroplasty on the clinical symptoms of the OA patients was not in question. Therefore, we consider it justified to assess responsiveness using change scores six months after TKA and THA for the construct validity approach. Additionally, for the criterion validity approach it is important that change can only be assumed in a portion of the study population. Therefore, we used the time point immediately after conservative treatment for this approach, in which only half of the study population received duloxetine. The direct effect of duloxetine was not the primary subject of the DOA study and its modest effect on OA pain has been established in previous studies [26, 27, 37–39].

In conclusion, this study endorses the responsiveness of the KOOS and HOOS pain subscales in patients with knee or hip OA and patients after TKA and THA based on construct and criterion validity approaches. The KOOS pain subscale might be able to detect the MIC at an individual level after arthroplasty, but both the KOOS and HOOS pain subscales were not able to do so after conservative treatment. Therefore, in clinical research, the study groups should be sufficiently large in order to reduce the measurement error when considering conservative treatment. This study is the first to report a considerable response shift in MIC of the KOOS and HOOS pain subscales. This should be taken into consideration when evaluating MIC of the KOOS and HOOS pain subscale after conservative versus operative treatment. Future research should present more reference data regarding MIC scores after different treatments.

## Supporting information

**S1 Data. Dataset underlying the results.**
(SAV)

## Acknowledgments

We thank Baukje Dijkstra, Wierd. P. Zijlstra, Tom M. van Raaij and Anita J. ten Hagen for their support in conducting the DOA-study.

## Author Contributions

**Conceptualization:** Wietske Rienstra, Martin Stevens, Tim Blikman, Sjoerd K. Bulstra, Inge van den Akker-Scheek.

**Data curation:** Wietske Rienstra, Tim Blikman.

**Formal analysis:** Wietske Rienstra, Inge van den Akker-Scheek.

**Funding acquisition:** Martin Stevens, Tim Blikman, Sjoerd K. Bulstra, Inge van den Akker-Scheek.

**Investigation:** Wietske Rienstra, Inge van den Akker-Scheek.

**Methodology:** Wietske Rienstra, Inge van den Akker-Scheek.

**Project administration:** Wietske Rienstra, Tim Blikman.

**Resources:** Wietske Rienstra, Sjoerd K. Bulstra.

**Software:** Wietske Rienstra.

**Supervision:** Martin Stevens, Sjoerd K. Bulstra, Inge van den Akker-Scheek.

**Validation:** Wietske Rienstra, Martin Stevens.

**Visualization:** Wietske Rienstra, Sjoerd K. Bulstra, Inge van den Akker-Scheek.

**Writing – original draft:** Wietske Rienstra.

**Writing – review & editing:** Wietske Rienstra, Martin Stevens, Tim Blikman, Sjoerd K. Bulstra, Inge van den Akker-Scheek.

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
