## [Decision Letter · Decision Letter 0]

2 Aug 2023

PONE-D-23-07675Responsiveness and Interpretability of the Pain subscale of the Knee and Hip Osteoarthritis Outcome Scale (KOOS and HOOS) in Osteoarthritis patients according to COSMIN guidelinesPLOS ONE

Dear Dr. Wietske Rienstra

Thank you for submitting your manuscript to PLOS ONE. After careful consideration, we feel that it has merit but does not fully meet PLOS ONE’s publication criteria as it currently stands. Therefore, we invite you to submit a revised version of the manuscript that addresses the points raised during the review process.

I would like to congratulate the authors for the well- written-manuscript. Some minor changes are requested.

Abstract

.    Line 16: “…osteoarthritis patients and after Total Knee Arthroplasty and Total Hip Arthroplasty (TKA/THA).”  should be corrected to “…osteoarthritis patients and evaluation of the results after Total Knee Arthroplasty (TKA) and Total Hip Arthroplasty (THA).”

.    The authors should give a small hint about the COSMIN guidelines in the “background” section in the “Abstract” (1-2 sentences)

.    Lines 21-22:  “This responsiveness study was conducted using data obtained from the DOA trial  (Duloxetine in OsteoArthritis).”  should be corrected to ““This responsiveness study was conducted using data obtained from the Duloxetine in OsteoArthritis (DOA) trial.” (the same in Lines 93 and 94)

.    Line 35: “93 participants” should be corrected to ‘Ninety- three participants”  

.    Lines 36-37-39: what is the meaning of “resp.”

.    Introduction

.    Line 67 “Total Knee Arthroplasty and Total Hip Arthroplasty (TKA/THA)” better to be changed to “Total Knee Arthroplasty (TKA) and Total Hip Arthroplasty (THA)

.    Line 90: “…subscales in patients with knee or hip OA as well as patients after TKA/THA” better to be changed to ““…subscales in patients with knee or hip OA as well as patients after TKA and THA” (the same for line 96)

.    Line 116: “HOOS/KOOS” should be corrected to Knee and Hip Osteoarthritis Outcome Scores (KOOS and HOOS)  

.    HOOS/KOOS  

.                 Lines 125-130: “Collins et al. report  that the Pain subscale especially, and also the ADL subscale, and QOL subscale of the KOOS show  the greatest relevance, room for improvement, and effect sizes in knee OA patients undergoing  TKA. Collins et al. report a Smallest Detectable Change (SDC) of 26 points on individual level for  the pain subscale of the KOOS in patients with OA[3]. To our knowledge, similar information is  lacking in literature regarding the HOOS.” These sentences are not a “Methodology” and should be moved to the “Discussion” section

.    Dutch Modified PainDETECT Questionnaire (mPDQ)  

.    Lines 142-144: “The Dutch version of the mPDQ is considered to be a fairly reliable and valid self-report  instrument in patients with hip and knee OA with an individual SDC of 7.3 points for hip/knee OA  patients[17,18]. This sentence is not a “Methodology” and should be moved to the “Discussion” section  

.    Line 150: “VAS have been reported as valid and reliable measures for the intensity of pain” this sentence should be deleted  

.    Line 248: The meaning of the abbreviation “BMI” was not previously mentioned.

We look forward to receiving your revised manuscript.

Kind regards,

Mohamed El-Sayed Abdel-Wanis, Ph.D.

Academic Editor

PLOS ONE

Journal Requirements:

Reviewers' comments:

Reviewer's Responses to Questions

**Comments to the Author**

1. Is the manuscript technically sound, and do the data support the conclusions?

Reviewer #1: Yes

Reviewer #2: Yes

2. Has the statistical analysis been performed appropriately and rigorously? 

Reviewer #1: Yes

Reviewer #2: Yes

3. Have the authors made all data underlying the findings in their manuscript fully available?

Reviewer #1: Yes

Reviewer #2: Yes

4. Is the manuscript presented in an intelligible fashion and written in standard English?

Reviewer #1: Yes

Reviewer #2: Yes

5. Review Comments to the Author

Reviewer #1: Overall, a well written manuscript.

Line 21 and line 93: Can the authors please explain what they mean by “responsiveness study”. Further clarity on this study design is needed as this is not clear in the manuscript and no evidence to support it was found in the literature. If this is an oversight, authors should kindly indicate the appropriate study design.

Line 245: Please consider reviewing this sentence as the phrase (see chapter 5 of this thesis) is out of place.

Line 253: A bit difficult to understand the table. Perhaps authors will consider repositioning the headers “Baseline”, “After conservative treatment” and “Six months after TKA/THA” for better presentation.

Thank you.

Reviewer #2: The authors assessed the responsiveness and interpretability of the subscale Pain of the patient-relevant outcome measures KOOS and HOOS in patients with osteoarthritis. The problem presented in the manuscript has not yet been investigated in the literature. This is a good paper with relevant discussion and conclusion. The paper was well written and succinct. I have, however, some minor concerns.

1. The overall perception of patient’s condition can be captured in a simple way with use of different instruments containing anchor questions. Why have you chosen the Global Impression Scale of Improvement?

2. The differences in the mean score changes in the Pain subscales between patients undergoing knee and hip arthroplasties results from the different time of recovery following these operations. It should be mentioned in the Discussion section.

3. In the Results section you mentioned that “reasons for loss to follow-up were described previously (…)” (line 245). It has not been done. As I understand this sentence was copied from the thesis (“see chapter 5 of this thesis”), and is not consistent with the present manuscript. Please, correct it. I would however like to see the data concerning loss to follow-up. This feature was not presented or discussed in your manuscript.

All in all, the manuscript is very interesting and written clearly in good English. It should be published after minor revisions.

6. PLOS authors have the option to publish the peer review history of their article (what does this mean?). If published, this will include your full peer review and any attached files.

Reviewer #1: **Yes: **Beatrice Efua Amoke Sankah

Reviewer #2: No

---

## [Author Response · Author response to Decision Letter 0]

17 Sep 2023

Dear Editor, 

Thank you for considering our manuscript for publication. We appreciate the time and effort the reviewers took to review the manuscript so extensively. 

Below, a point-by-point response is provided to all the peer reviewers’ comments and the comments from the Academic Editor. Also, the additional requirements of the journal are addressed below.

We hope that with these revisions, the manuscript will fully meet PLOS ONE’s publication criteria.

Kind regards on behalf of all the authors,

Wietske Rienstra

Comments by the academic editor:

Line 16: “…osteoarthritis patients and after Total Knee Arthroplasty and Total Hip Arthroplasty (TKA/THA).”  should be corrected to “…osteoarthritis patients and evaluation of the results after Total Knee Arthroplasty (TKA) and Total Hip Arthroplasty (THA).” 

done

The authors should give a small hint about the COSMIN guidelines in the “background” section in the “Abstract” (1-2 sentences)

See the additional sentences in the manuscript (Line 21-24)

Lines 21-22: “This responsiveness study was conducted using data obtained from the DOA trial  (Duloxetine in OsteoArthritis).”  should be corrected to ““This responsiveness study was conducted using data obtained from the Duloxetine in OsteoArthritis (DOA) trial.” (the same in Lines 93 and 94) 

done

Line 35: “93 participants” should be corrected to ‘Ninety- three participants”   

done

Lines 36-37-39: what is the meaning of “resp.” 

The meaning is ‘ respectively’. For clarification I corrected the abbreviation into the complete word in the manuscript

Line 67 “Total Knee Arthroplasty and Total Hip Arthroplasty (TKA/THA)” better to be changed to “Total Knee Arthroplasty (TKA) and Total Hip Arthroplasty (THA) 

Thank you for this suggestion, see the correction in the manuscript

Line 90: “…subscales in patients with knee or hip OA as well as patients after TKA/THA” better to be changed to ““…subscales in patients with knee or hip OA as well as patients after TKA and THA” (the same for line 96) 

Thank you for this suggestion, TKA/THA was changed into TKA and THA throughout the entire manuscript

Line 116: “HOOS/KOOS” should be corrected to Knee and Hip Osteoarthritis Outcome Scores (KOOS and HOOS)   done

HOOS/KOOS Changed into KOOS and HOOS throughout the entire manuscript

Lines 125-130: “Collins et al. report  that the Pain subscale especially, and also the ADL subscale, and QOL subscale of the KOOS show  the greatest relevance, room for improvement, and effect sizes in knee OA patients undergoing  TKA. Collins et al. report a Smallest Detectable Change (SDC) of 26 points on individual level for  the pain subscale of the KOOS in patients with OA[3]. To our knowledge, similar information is  lacking in literature regarding the HOOS.” These sentences are not a “Methodology” and should be moved to the “Discussion” section 

This paragraph was moved to the discussion section

Lines 142-144: “The Dutch version of the mPDQ is considered to be a fairly reliable and valid self-report  instrument in patients with hip and knee OA with an individual SDC of 7.3 points for hip/knee OA  patients[17,18]. This sentence is not a “Methodology” and should be moved to the “Discussion” section 

This paragraph was moved to the discussion section

Line 150: “VAS have been reported as valid and reliable measures for the intensity of pain” this sentence should be deleted   

This sentence was deleted

Line 248: The meaning of the abbreviation “BMI” was not previously mentioned. 

The abbreviation was clarified

Comments by Reviewer #1:

Line 21 and line 93: Can the authors please explain what they mean by “responsiveness study”. Further clarity on this study design is needed as this is not clear in the manuscript and no evidence to support it was found in the literature. If this is an oversight, authors should kindly indicate the appropriate study design. 

Thank you for this comment. The study design was clarified into: clinimetric study 

Line 245: Please consider reviewing this sentence as the phrase (see chapter 5 of this thesis) is out of place. T

This sentence was reviewed. The mentioned 'chapter 5' has been changed into the proper reference (nowadays the paper has been published) (Rienstra et al. 2021. BMJ Open)

Line 253: A bit difficult to understand the table. Perhaps authors will consider repositioning the headers “Baseline”, “After conservative treatment” and “Six months after TKA/THA” for better presentation.

The presentation of the table was changed with the aim of improving readability

Comments by Reviewer #2:

The overall perception of patient’s condition can be captured in a simple way with use of different instruments containing anchor questions. Why have you chosen the Global Impression Scale of Improvement?

Thank you kindly for this suggestion. In the period in which the trial was conceived we were unaware of such instruments. Therefore we used the Global Impression Scale of Improvement as an anchor.

The differences in the mean score changes in the Pain subscales between patients undergoing knee and hip arthroplasties results from the different time of recovery following these operations. It should be mentioned in the Discussion section.

Thank you for this very true comment. However, since a clinimetric study as this regards whether the measured change scores are valid rather than the magnitude of the changes (or the true difference between hip and knee arthroplasty patients), this comment is more relevant for discussion in the paper regarding the primary trial results (Rienstra et al. 2021. BMJ Open). Therefore we chose to discard this comment in order to keep to the clinimetric character of this paper.

In the Results section you mentioned that “reasons for loss to follow-up were described previously (…)” (line 245). It has not been done. As I understand this sentence was copied from the thesis (“see chapter 5 of this thesis”), and is not consistent with the present manuscript. Please, correct it. 

See also the reply to reviewer #1 regarding this issue. The appropriate reference was added.

I would however like to see the data concerning loss to follow-up. This feature was not presented or discussed in your manuscript. 

As this information is provided in a free access paper we chose not to elaborate on this part for multiple reasons:

1. To avoid repeating of identical results and information in different papers

2. Because we think that the context of this information is better described and more relevant in the paper regarding the original trial

3. To improve readability of the present manuscript.

Journal Requirements:

https://journals.plos.org/plosone/s/file?id=wjVg/PLOSOne_formatting_sample_main_body.pdfand

-We have updated the style according to the abovementioned requirements.

We have included the dataset underlying the results as a supplementary file (corrected the file name to make it clearer). 

-The location of the underlying dataset in the supplementary file is mentioned in line 299.

-We have included these.

-The reference list was updated

---

## [Decision Letter · Decision Letter 1]

19 Oct 2023

Responsiveness and Interpretability of the Pain subscale of the Knee and Hip Osteoarthritis Outcome Scale (KOOS and HOOS) in Osteoarthritis patients according to COSMIN guidelines

PONE-D-23-07675R1

Dear Dr. Wietske Rienstra 

We’re pleased to inform you that your manuscript has been judged scientifically suitable for publication and will be formally accepted for publication once it meets all outstanding technical requirements.

Kind regards,

Mohamed El-Sayed Abdel-Wanis, Ph.D.

Academic Editor

PLOS ONE

Additional Editor Comments (optional):

The authors addressed all the reviewer and editorial comments

Reviewers' comments:

Reviewer's Responses to Questions

**Comments to the Author**

1. If the authors have adequately addressed your comments raised in a previous round of review and you feel that this manuscript is now acceptable for publication, you may indicate that here to bypass the “Comments to the Author” section, enter your conflict of interest statement in the “Confidential to Editor” section, and submit your "Accept" recommendation.

Reviewer #1: All comments have been addressed

2. Is the manuscript technically sound, and do the data support the conclusions?

Reviewer #1: (No Response)

3. Has the statistical analysis been performed appropriately and rigorously? 

Reviewer #1: (No Response)

4. Have the authors made all data underlying the findings in their manuscript fully available?

Reviewer #1: (No Response)

5. Is the manuscript presented in an intelligible fashion and written in standard English?

Reviewer #1: (No Response)

6. Review Comments to the Author

Reviewer #1: (No Response)

7. PLOS authors have the option to publish the peer review history of their article (what does this mean?). If published, this will include your full peer review and any attached files.

Reviewer #1: **Yes: **Beatrice Efua Amoke Sankah

---

## [Editor Report · Acceptance letter]

6 Nov 2023

PONE-D-23-07675R1 

Responsiveness and Interpretability of the Pain subscale of the Knee and Hip Osteoarthritis Outcome Scale (KOOS and HOOS) in Osteoarthritis patients according to COSMIN guidelines 

Dear Dr. Rienstra:

I'm pleased to inform you that your manuscript has been deemed suitable for publication in PLOS ONE. Congratulations! Your manuscript is now with our production department. 

Kind regards, 

on behalf of

Prof. Dr Mohamed El-Sayed Abdel-Wanis 

Academic Editor

PLOS ONE